

# Wind farm power production assessment: a comparative analysis of two actuator disc methods and two analytical wake models

Nikolaos Simisiroglou[1,2], Heracles Polatidis[2], and Stefan Ivanell[2]

[1]WindSim AS, Fjordgaten 15, N-3125 Tønsberg , Norway
[2]Uppsala University, Wind Energy Section, Campus Gotland, 621 67 Visby, Sweden

*Correspondence to:* Nikolaos Simisiroglou (nikolaos.simisiroglou@geo.uu.se)

**Abstract.** The aim of the present study is to perform a comparative analysis of two actuator disc methods (ACD) and two analytical wake models for wind farm power production assessment. To do so wind turbine power production data from the Lillgrund offshore wind farm in Sweden is used. The measured power production for individual wind turbines is compared with results from simulations, done in the WindSim software, using two ACD methods (old and new) and two analytical wake
models widely used within the wind industry (Jensen and Larsen wake models). It was found that the new ACD method and the Larsen model outperform the other method and model in most cases. Furthermore, results from the new ACD method show a clear improvement in the estimated power production in comparison to the old ACD method. The Jensen method seems to overestimate the power deficit for all cases. The new ACD method, despite it's simplicity, is capable of capturing the power production within the given error margin although it tends to underestimate the power deficit.

## 1 Introduction

As a wind turbine extracts energy from the wind it creates a region downstream where the wind velocity is decreased and the turbulence intensity is increased. This region is commonly called the wake region and represents the effect of the wind turbine on the free flow of the wind. Turbines are currently placed in close configurations for economic reasons. The result of these configurations are that turbines often operate in the wake of other turbines which results in reduced production. In the context of
an offshore wind farm this power loss is in the order of 5% to 20% (Barthelmie et al. (2009)). It is thus apparent that the ability to accurately predict wind turbine wakes has a significant impact on increasing wind farm profitability. This insight can provide value in at least two stages of a wind farm's project lifespan, i.e. the pre–construction phase and the operational phase. In the pre–construction phase the value is added by being able to calculate with higher accuracy the wind flow conditions within the planned wind farm and layouts that decrease wake losses and turbine loads may be designed for. During the operational phase
wind farm managers in many cases sacrifice power production in favour of decreasing wind flow related loads to extend the lifetime of the wind turbines. Subsequently, by accurately modelling wind farm wakes, wind farm managers could create better controller strategies per inflow condition that may increase the power production without negatively impacting the operational lifetime of the wind farm.



Traditionally the impact of wakes is accounted for by using analytical wake models such as the Jensen (1983) and the Larsen (1988) models. Even though these models are extremely fast in computing the wake deficit of an entire wind farm, they depend on coefficients that need to be empirically determined per case, making them not generically applicable to all sites. An example of one such empirical variable is the wake expansion coefficient of the Jensen model, presented later in Section 2.2, Fig. 2.

Furthermore they also suffer from intrinsic simplifications when calculating, for instance, the wake to wake interaction and the turbulence characteristics of the wake (Katic et al. (1987); Troldborg (2009)). Very recently Seim et al. (2017) validated three wind turbine analytical wake models in a complex terrain context using wind farm production data.

More advanced computational fluid dynamics (CFD) techniques to assess wind turbine wakes are performed by modelling the wind turbine forces in a simplification of the Navier-Stokes equation e.g. Large Eddy Simulations (LES). Two widely used

methods to model the wind turbine forces are the actuator line (ACL) method and the actuator disc (ACD) methods. The ACL method, proposed by Sørensen and Shen (2002), involves distributing the calculated forces along rotating lines that represent the blades of an actual wind turbine. Research with LES and the ACL method has been performed by e.g. Ivanell et al. (2007); Nilsson et al. (2015b); Sarmast et al. (2014). The ACD method, in which the wind turbine rotor is represented by distributing the forces over a porous disc, has been investigated with LES by e.g. Breton et al. (2014); Wu and Porté-Agel (2011); Olivares-

Espinosa et al. (2014); Laan et al. (2015); Nilsson et al. (2015a). Although LES provide high fidelity results compared to field measurements, the computational requirements of the method are far too high and therefore not currently suitable for the engineering requirement of computing wakes for an entire wind farm (Churchfield et al. (2012); Laan et al. (2015)). A less computationally expensive alternative to LES are the Reynolds averaged Navier-Stokes simulations (RANS). Here the effects of the turbulent eddies are determined by turbulent models without resolving them in detail and in which steady state averaged

results are computed. RANS simulations have been used with the ACD method to simulate wind turbine wakes by numerous researchers, e.g. Laan et al. (2015); Castellani and Vignaroli (2013); Prospathopoulos et al. (2011); El Kasmi and Masson (2008); Sumner et al. (2013).

In previous work from Crasto and Gravdahl (2008) an ACD method is introduced for modelling wakes in the wind farm development software WindSim AS (2017), herein referred to as the old ACD method. More recently Simisiroglou et al. (2016)

developed a new ACD method and validated it against three different wind tunnel test cases, this method herein referred to as the new ACD method. In this paper a comparative analysis of these two ACD methods and the Jensen and Larsen analytical wake models is performed in the context of a case study. This is done by comparing power production measurement data from the Lillgrund offshore wind farm located in Øresund, between Sweden and Denmark with results from the RANS simulations.

The paper unfolds as follows: section 2 introduces the methodological framework and shortly presents the two ACD methods,

the analytical wake models and the boundary conditions of the numerical set–up of the study. Section 3 presents the offshore wind farm Lillgrund, where the data are taken from and the inflow directions used for the comparative analysis. In section 4 the results are presented and discussed. Lastly, in section 5 the main conclusions of this study are drawn and proposals for further research are highlighted.





**Figure 1.** Methodological framework.

## 2   Methodological Framework

Fig. 1 presents the methodological framework of the study. The wind farm power production is found by taking into account wind turbine wake losses. The wake losses can be assessed, among other ways, by using the analytical wake models or the ACD method. Herein two ACD methods and two analytical wake models are used to assess the power production through simulations. The results from the simulations are compared with the power production data from the offshore wind farm Lillgrund. In the remaining portion of this section the two ACD methods will be introduced along with the two analytical wake models and the boundary conditions used to set–up the simulations of the study.

### 2.1   ACD method

The ACD method is a way to represent the wind turbine's effect on the wind flow in a simulation. To do so, a thrust force calculated from the 1D momentum theory is applied to a porous disc, which in turn acts as a momentum sink.



For the new ACD method presented in Simisiroglou et al. (2016) the thrust force $F_i$ at each cell of the disc is calculated from

$$F_i = C_T\left(U_{1,i}\right) \frac{1}{2}\rho \left(\frac{U_{1,i}}{1-\alpha_i}\right)^2 A_i \tag{1}$$

Where $U_{1,i}$ is the velocity of the flow at $i$-th cell of the disc, $\alpha_i$ is the axial induction factor calculated for each individual cell of the disc, $A_i$ is the surface area of the cell facing the undisturbed wind flow direction and $C_T\left(U_{1,i}\right)$ is a modified thrust coefficient dependent on the velocity at the disc $U_{1,i}$ and $\rho$ is the air density set to 1.225 kg m$^{-3}$. In most cases wind turbine manufacturers offer $C_T$ as a function of $U_\infty$, the undistributed wind velocity. This $C_T$ is reasonable for the first wind turbine of the row but not for the downstream wind turbines were the flow has been disturbed. Hence, in the present case a $C_T$ which is a function of $U_1$, the velocity of the disc, is needed. This function can be established from the 1D momentum theory by combining the definition of the trust coefficient $C_T$ and the axial induction factor $\alpha$

$$C_T = 4\alpha(1-\alpha) \tag{2}$$
$$U_1 = (1-\alpha)U_\infty \tag{3}$$

Hence from Eq. 2 and Eq. 3 the following is obtained

$$U_1 = U_\infty \left(1 - \frac{1}{2}\left(1 - \sqrt{1 - C_T\left(U_\infty\right)}\right)\right) \tag{4}$$

The power production is estimated by finding the average induction factor over the disc as $\overline{a} = \sum a_i$, and then for each velocity $U_{1,i}$ over the disc an undisturbed wind velocity $U_{\infty,i}$ is found using the following equation

$$U_{\infty,i} = \frac{U_{1,i}}{1-\overline{a}} \tag{5}$$

For each undisturbed wind velocity a power is found using the power curve and then they are averaged over the disc. The data for the thrust coefficient curve $C_T\left(U_\infty\right)$ is supplied by Hansen (2013) and the data for the power curve is provided by Jeppsson et al. (2008).

The main difference between the new ACD method and the old ACD method presented in Crasto and Gravdahl (2008), is that the latter utilizes a uniform thrust distribution instead of calculating the thrust at each cell and the power production of the wind turbine is assessed solely from the velocity $U_1$ at hub height instead of the entire rotor.

## 2.2 Analytical wake models

Two analytical wake models are used in this study, the Jensen (1983) and the Larsen (2009). The Jensen model is based on the description of a single wake behind a rotor. Here the normalised wind velocity deficit $\delta V = \frac{U_\infty - V}{U_\infty}$, at a distance $x$ behind a single wind turbine with a thrust coefficient $C_T$ is found by

$$\delta V = \frac{1 - \sqrt{1 - C_T}}{\left(1 + \frac{2kx}{D}\right)^2} \tag{6}$$



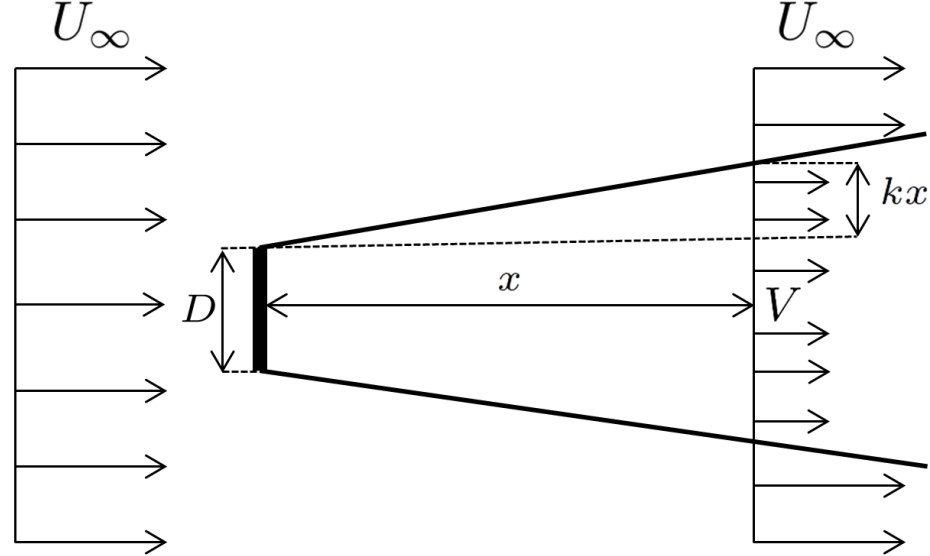

**Figure 2.** Schematic representation of the Jensen wake model. Where $D$ is the rotor diameter, $U_\infty$ the undisturbed wind velocity, $x$ the downstream distance of the wake from the rotor, $V$ the velocity within the wake and $k$ the wake expansion coefficient.

Where $k$ is the wake expansion coefficient, $D$ the rotor diameter of the wind turbine and $V$ the wind velocity in the wake at position $x$ (Fig.2). When multiple wakes influence the velocity at a position, the total normalised velocity deficit $V_{tot.}$, is found by

$$\delta V_{tot.} = \sqrt{\sum \delta V_j^2} \tag{7}$$

Where $\delta V_j$ is the normalised wind velocity deficit from the j-th wind turbine.

The Larsen model is based on turbulent boundary equations and a similarity assumption. By neglecting different terms in the governing equations, two different versions of the model are presented in Larsen (1988). Here we will be using the first order wake model in which the normalised velocity deficit is described by

$$\delta V = \frac{\left(C_T A x^{-2}\right)^{1/3}}{9} \left[ r_x^{3/2} \left(3c_1^2 C_T A x\right)^{-1/2} - \frac{35}{2\pi}^{3/10} \left(3c_1^2\right)^{-1/5} \right]^2 \tag{8}$$

Were $r_x$ is the radial distance at a position $x$ downstream of the rotor and $c_1$ is a constant. A notable difference between the Jensen and Larsen model is that the wake deficit for the Larsen model is not uniform as is the case for the Jensen model, but differs radially in the cross-section of the wake, due to $r_x$. The effect of multiple wakes on the velocity field at a position is also taken into account by Eq. 7.

The power production at each wind turbine is found by the velocity at hub height minus the velocity deficit calculated by
either the Jensen or Larsen wake model.



## 2.3 Boundary conditions

The simulations are performed by the commercial CFD code WindSim. The RANS equations are solved via a general collo­cated velocity method (GCV) (Semin et al. (1996)) in a body fitted coordinate grid formulation (BFC). Therefore, the variables are stored on a collocated mesh. The GCV method, however, uses a segregated pressure–based solver strategy, thus a momen­tum interpolation algorithm is performed to evaluate mass fluxes on the control volume faces. This algorithm is a variant of the Rhie–Chow momentum interpolation method (Rhie (1982); Rhie and Chow (1983)), which avoids the undesired side effect of producing solutions that depend on the relaxation factors used (Majumdar (1988)). The $k - \varepsilon$ two equation turbulence model of Launder and Spalding (1974) is used to close the equations. The hybrid method of Spalding (1972) is used to discretize the convective terms. The diffusion terms, on the other hand, are discretized by using the central differencing scheme. At the inlet of the domain, a log law velocity profile is set with the turbulence parameters obeying the following equations

$$U(z) = \frac{U_i^*}{\kappa} \ln\left(\frac{z}{z_0}\right), \qquad k = \frac{U_i^{*2}}{0.3} \qquad \text{and} \qquad \varepsilon = \frac{U_i^{*3}}{\kappa z} \tag{9}$$

Where $U(z)$ is the streamwise wind velocity at height $z$, $U_i^*$ is the inlet friction velocity, $\kappa$ is von Kármán's constant set to 0.41, $z_0$ is the effective roughness height, $k$ is the turbulent kinetic energy and $\varepsilon$ is the dissipation rate. The ambient turbulence intensity at hub height ($TI_h$) assuming a isotropic normal stress approximation is given by

$$TI_h = \frac{\sqrt{\frac{2}{3}k}}{U_h} \Rightarrow TI_h = \frac{\kappa\sqrt{\frac{2}{3}}}{\ln\left(\frac{z_h}{z_0}\right)\sqrt[4]{C_\mu}} \tag{10}$$

Where $U_h$ is the streamwise wind velocity at hub height $z_h$ and $C_\mu$ is an empirical constant equal to 0.09. This value for $C_\mu$ was recommended by Launder et al. (1973) after researching free turbulent flows. By knowing the values for the turbulence intensity at hub height (see Table 1, in Section 3), using Eq. (10) the appropriate value for the effective roughness height ($z_0$) is found. The wall function of the ground surface is described by the following equations

$$U_r = \frac{U_w^*}{\kappa} \ln\frac{z_r}{z_0}, \qquad k = \frac{U_w^{*2}}{0.3} \qquad \text{and} \qquad \varepsilon = \frac{C_\mu^{0.75} k^{1.5}}{\kappa z_r} \tag{11}$$

Where $U_r$ is the absolute value of the velocity parallel to the wall at the first grid node and $z_r$ is the normal distance of the first grid node from the wall. Here the wall friction velocity $U_w^*$ is calculated from $U_w^* = \sqrt{\tau_w/\rho}$, the wall shear stress is $\tau_w = s\rho U_r^2$ and $s = \left(\frac{\kappa}{\ln\frac{z_r}{z_0}}\right)^2$. The lateral walls of the domain are impermeable and frictionless. The outlet and top plane are treated as a fixed pressure boundary. Moreover, a diffusive link is set at the top plane to help preserve the inlet boundary profile calculated via Eq. (9) (CHAM (2017)). The pressure value at the outlet is set to zero, which is equal to an atmospheric pressure of 101.3250 kPa.

## 3 Lillgrund offshore wind farm

The total rated power of the Lillgrund wind farm is 110.4 MW which consists of 48 Siemens SWT-2.9-93 wind turbines with a rated power of 2.3 MW each. The rotor diameter of the Siemens SWT-2.9-93 wind turbine is 92.6 m and the hub height is at 65



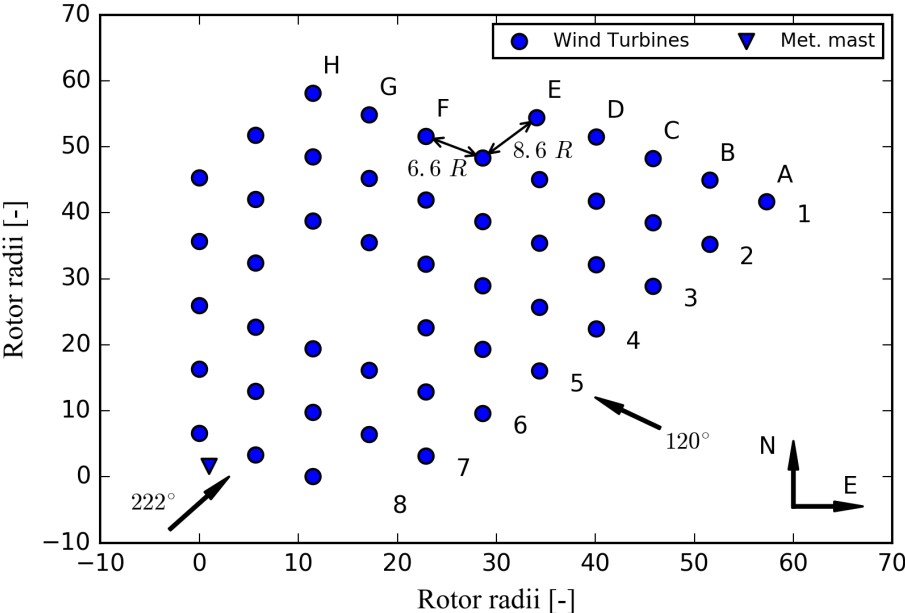

**Figure 3.** Layout of Lillgrund wind farm, where $R$ is the rotor radius. $120°$ and $222°$ are the inflow directions to be investigated.

m. The wind farm layout is shown in Fig. 3. The convention used herein to label each individual wind turbine is to assign them a letter and number, the letter represents the column in which the wind turbine is located ranging from A-H and the number represents the appointed row ranging from 1-8. A distinct feature of the Lillgrund wind farm is the very tight configuration of the wind turbines, with a separation distance of $6.6$ rotor radii (R), for the columns and $8.6R$ for the rows. The reason behind

this unusually compact inter–row spacing according to Dahlberg (2009) is that the layout was initially developed for a smaller wind turbine model. This model was however unavailable when the construction permit for the wind farm was finally given. As a result the developers decided to use a larger wind turbine while keeping the original layout. Another noteworthy feature of this layout is the gap in the middle of the farm where wind turbines D5 and E5 should have been placed. This gap increases the separation distance locally and will have an effect on the total wake recovery and power production of the wind turbines

downstream of the gap.

The turbulence intensity per direction used in this study was estimated by data extracted from a met mast, located to the south–west as seen in Fig. 3, during approximately a two year period prior to the erection of the wind turbines ($1^{st.}$ of September 2013 to the $28^{th.}$ of February 2016) Bergström (2009). The inflow directions on which this study will focus are shown in Table 1 along with the respective turbulence intensities at a 65 m height. Specifically, for the wind flow direction of $120°$, data are

used from Rows 3 and 5 at a wind velocity of 9 m s$^{-1}$. Similarly, for the $222°$ inflow direction, data are used from column B and D at a wind velocity of 9 m s$^{-1}$. Rows 3,5 and columns B,D have been selected because they represent two distinct cases in which one row/column includes the gap and the other does not (Fig 3).



Records from the wind turbines' SCADA system are available for 10 minute time periods. The power production of each wind turbine per inflow direction and undisturbed wind speed is thus available. Due to data availability limitations the power production data are binned based on inflow directions and wind velocity measurements. The directional bins are $\pm 2.5°$ wide and the velocity bins are $\pm 0.5$ m s$^{-1}$ wide as seen in Table 1 below.

**Table 1.** Main inflow directions and information.

| Description | Inflow direction [degrees] | Row/Column | Wind Velocity [m s$^{-1}$] | Turbulence Intensity (%) |
|---|---|---|---|---|
| Southeast | $120 \pm 2.5°$ | 3, 5 | $9.0 \pm 0.5$ | 5.5 |
| Southwest | $222 \pm 2.5°$ | B, D | $9.0 \pm 0.5$ | 5.6 |

The inflow wind direction is calculated by the average wind turbine yaw position from a group of wind turbines. For the $120°$ direction, the group of wind turbines consists of turbines A1 to A7; similarly for the $222°$ direction the wind turbines B8 to D8 are used. The undisturbed wind velocity used per direction is derived by averaging the undisturbed wind velocity of all wind turbines in the noted groups. The undisturbed velocity of each wind turbine in the group is found by comparing the 10 minute power production of the turbine to the official power curve as found in Jeppsson et al. (2008). It should be noted, that
for these data a filtration depending on the atmospheric stability condition has not been performed.

## 4 Results and Discussion

Results of the comparative analysis will be presented in this section after normalization. The normalization is done by dividing the power production of each wind turbine of the row/column with the measured power production of the first wind turbine in the row/column. The reported error bar range is equal to the normalised standard deviation of the measured power production.

Figure 4 below presents the results of Row 3 and 5 for the $120°$ wind direction. It is observed that the power production of the first wind turbine is captured by all methods/models. Regarding the second wind turbine of the row, however, the new ACD method and the Larsen model do not capture the steep reduction of the measured power output. In contrast, the old ACD method and the Jensen model give results within the error range. The power production of the third wind turbine is captured within the error range when using the new ACD and the Larsen model, which is not the case for the old ACD and the Jensen model.

None of the methods/models are able to capture the sharp power production increase observed in the measurements from the second to the third wind turbine. The power production of the subsequent wind turbines in Fig. 4 (a) is captured within good accuracy when using the new ACD method, whereas the other method/models underestimate the power production. Regarding Fig. 4 (b), the increase of the power output observed between the C5 and F5 wind turbines due to the gap, is captured best by the new ACD method.

Other researchers have also studied the Lillgrund wind farm case. van der Laan et al. (2015) used the $k - \varepsilon$ and the $k - \varepsilon$ $f_p$ turbulence models to investigate their effects on the simulated power production for rows 3 and 5. Their results with the

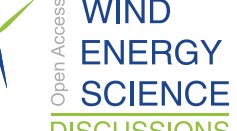

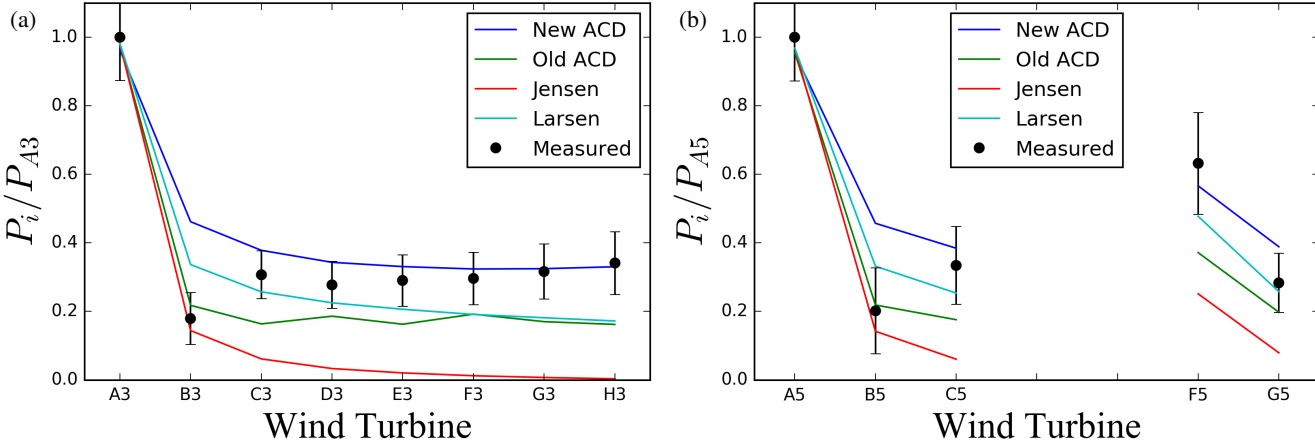

**Figure 4.** Normalised power production for the $120°$ direction (a) Row 3 and (b) Row 5.

$k - \varepsilon$ turbulence model coincide with the ones presented here for the new ACD method case. On the other hand, the $k - \varepsilon \, f_p$ made better predictions of the power production, especially for the second wind turbine in the row. Furthermore, Nilsson et al. (2015a) have used LES with an ACD method to simulate power production for the same wind turbines rows of the Lillgrund wind farm. Their simulations predicted with better accuracy the power production of the same wind turbine rows including the

5 power deficit of the second wind turbine and the power increase between the C5 and the F5 wind turbines. That is expected since LES fully resolves the effect of the large turbulent eddies.

Results of columns B and D for the $222°$ wind direction case are presented in Fig 5.

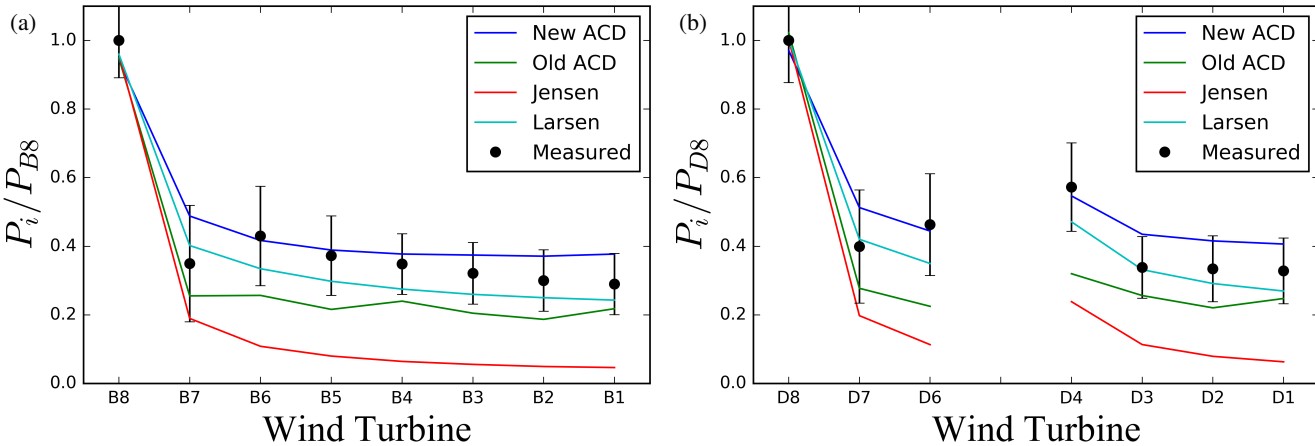

**Figure 5.** Normalised power production for the $222°$ direction (a) Column B and (b) Column D.





Here again the new ACD method and the Larsen model over–predict the power production of the second wind turbine, whereas the old ACD and the Jensen model under–predict the power production. For the rest of the wind turbines of the rows the new ACD method and the Larsen model outperform the old ACD method and the Jensen model.

van der Laan et al. (2015) obtained results similar to those of the new ACD method presented here when using the $k - \varepsilon$
turbulence model. In addition, they showed that their results improved with the $k - \varepsilon \ f_p$ turbulence model. Moreover, LES with an ACD and ACL method have been performed by Nilsson et al. (2015a) and Churchfield et al. (2012) respectively, with quite good accuracy. Nilsson et al. (2015a) have generally slightly over predicted the measurements whereas results of Churchfield et al. (2012) vary depending on the particular wind turbine examined.

While this study has provided a clear comparison between the new and old actuator disc methods and two analytical wake models with experimental data, it is also important to mention its limitations:

- When presenting the results, the directional uncertainty as proposed from Gaumond et al. (2014) has not been considered.

- Yaw misalignment errors are not considered.

- The data has not been filtered for stability.

- The study has been performed only for cases where the wind direction is in–line to the row of wind turbines.

- Only the $k - \varepsilon$ turbulence model is used.

- In the simulations one power curve is used; in reality depending on the environmental conditions e.g. turbulence intensity or air density, wind turbines operate on a range of power curves.

In spite of these limitations, the present study has presented a straightforward comparison among different methods to estimate the power production of a wind farm. Even though the ACDs used in this study are both based on the 1D momentum theory, it is shown that they provide different results as a consequence of how the forces are calculated and distributed over the disc and how the power is estimated. The new ACD method and the Larsen model, in most cases, seem to better estimate the power production in comparison to the other method and model. The Jensen model in particular seems in general to greatly underestimate the power production of the downwind turbines.

## 5 Conclusions

This paper compared the power production results of four different methods/models against measurements for the offshore wind farm of Lillgrund, which is located in Sweden. Two ACD methods based on the 1D momentum theory are compared along with two analytical wake models, the Jensen and the Larsen. For this comparison two main wind directions are investigated. The main conclusions are: (i) The new ACD method and the Larsen model outperform the other method and model in most cases. (ii) The power increase of the turbine after the gap is better captured when using the new ACD method. (iii) The results from the new ACD method show a clear improvement in the estimated power production in comparison to the old ACD method.



(iv) The Jensen method seems to overestimate the power deficit for all cases. (iv) The new ACD method, despite its simplicity, is capable of capturing the power production within the error margin although it tends to underestimate the power deficit. One may say that the new ACD method in RANS, which has much lower computational requirements than the ACD method in LES at the cost of lower accuracy, could represent a good compromise.

van der Laan et al. (2015) showed that by using the $k - \varepsilon \ f_p$ turbulence model the power production results for the second wind turbine in the row are in closer agreement with the measurements than when using the $k - \varepsilon$ model. Hence future work could focus on researching the impact of using different turbulence closure models on the results. Furthermore, as the Lillgrund wind farm is quite unique due to its very close inter-row spacing of the wind turbines, it would be advantageous to apply a similar comparative analysis using these methods/models on wind farms with larger distances between wind turbines.

Furthermore, as stability has an impact on wake development it is important to explore wind farm cases in which the data can be filtered for stability. Finally, it would prove beneficial to continue investigating the offshore Lillgrund wind farm for other wind flow conditions to obtain a more comprehensive understanding of the performance of the methods/models.

*Acknowledgements.* This work is financially supported by the Research Council of Norway (Project no. 231831). The filtration and validation of the data was performed by Kurt Schaldemose Hansen (DTU Institute for Vindenergy Denmark) according to the guidelines described in

Barthelmie et al. (2011). Andrew Barney is kindly acknowledged for proof reading the manuscript.





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
