# Peer review of "Wind farm power production assessment: a comparative analysis of two actuator disc methods and two analytical wake models"

_Wind Energy Science, 2018_

## Referee Comment (RC1) · Anonymous Referee #1 · 16 Mar 2018

The submitted manuscript is quite interesting and well written but I think that must be improved in some key point of the discussion. The main outcome of the paper spreads from the comparison of different wake models: two of them are empirical/engineering models with a quite different approach while the other two actuator disc models seems quite similar and the reader is not able to understand the main reasons for such different results. To improve the "scientific soundness" of the paper a deeper discussion on the differences between the new and old ACD model is strongly recommended. About the new ACD two things need to be better explained: 1. If the equation for the thrust is really different from the old version and how 2. How it is performed the estimation of the power output (please explain

equation 5) and why you used an average induction to obtain a local undisturbed wind. Beside this I think that there is also a mistake in the text: in the old version of the ACD there was also already the possibility to have different force distributions and not only the uniform one (this is quite clear from the WindSim web site in the section of the presentation, see the slides from G. Crasto at the usermeeting 2011 http://windsim.com/documentation/UM2011pres/1106_WindSim_UM_WS_Crasto.pdf).

---

## Referee Comment (RC2) · Anonymous Referee #2 · 27 Apr 2018

The paper "Wind farm power production assessment: a comparative analysis of two actuator disc methods and two analytical wake models" by Simisiroglou et al. Describes a comparison of RANS – wake simulations and analytical wake models with Power data from the Lillgrund offshore wind farm. The authors introduce a new method to calculate the thrust coefficient for the actuator disc model in RANS, that calculates lower wake losses than the method that the authors consider as standard. I would not recommend a publication in Wind Energy Science for following reasons:

1) The study offers little new content. The Lillgrund data has been used as validation data for wake models already in e.g. Gaumond et al, 2012: "Benchmarking of wind

turbine wake models in large offshore wind farms" (also with Larsen and Jensen model) and Keck et al, 2014 "Validation of the standalone implementation of the dynamic wake meandering model for power production".

2) The authors compare their "new" actuator disc approach with a very simplified version of an actuator disc method. The simplification of using only hub height wind speed is not the current standard. E.g. van der Laan et al, 2015 "The k-epsilon-fp model applied to double wind turbine wakes using different actuator disk force methods" or Wu and Porte-Agel, 2011: "Large-eddy simulation of wind-turbine wakes: evaluation of turbine parametrisations" use at least an average of the wind speed over the rotor to calculate the free-stream velocity.

3) The authors use the standard k-epsilon model for the turbulence closure in RANS. Van der Laan et al, 2015 "An improved k-eps model applied to wind turbine wake in atmospheric turbulence "and Rethoré, 2009 "Wind turbine wake in atmospheric turbulence" have shown with wind measurements and LES that this model is not capable of replicating wakes of isolated wind turbines. The change of the thrust coefficient does not change this behavior.

I could see a value in the contribution, if the authors focus on the RANS calculation and do more literature research on the state of art of actuator disc and turbulence modeling and use these approaches as comparison.

Further comments:

- The description of the methods is incomplete. The coefficients used for the analytical models are missing. There is no information about the mesh of the RANS calculations.
- The figures should be readable in grey-scale.

---

## Author Comment (AC1) · 23 May 2018

Dear Referee,

We would like to thank you for putting time and effort to provide for a thorough and detailed review of our manuscript.

Please find attached a copy of our revised manuscript and a point-to-point in the form of a supplement.

We hope that we have now addressed all comments raised and the manuscript has reached adequate quality to warrant publication.

[Figure]

Yours sincerely,

Nikolaos Simisiroglou

Please also note the supplement to this comment:
https://www.wind-energ-sci-discuss.net/wes-2018-8/wes-2018-8-AC1-supplement.zip
* * *